# Subacute Effects of Moderate-Intensity Aerobic Exercise in the Fasted State on Cell Metabolism and Signaling in Sedentary Rats

**DOI:** 10.3390/nu16203529

**Published:** 2024-10-18

**Authors:** Layane Ramos Ayres, Éverton Lopes Vogt, Helena Trevisan Schroeder, Mariana Kras Borges Russo, Maiza Cristina Von Dentz, Débora Santos Rocha, Jorge Felipe Argenta Model, Lucas Stahlhöfer Kowalewski, Samir Khal de Souza, Vitória de Oliveira Girelli, Jerônimo da Rosa Coelho, Nathalia de Souza Vargas, Alvaro Reischak-Oliveira, Paulo Ivo Homem de Bittencourt, Eurico Nestor Wilhelm, Anapaula Sommer Vinagre, Mauricio Krause

**Affiliations:** 1Laboratório de Inflamação, Metabolismo e Exercício (LAPIMEX) e Laboratório de Fisiologia Celular, Departamento de Fisiologia, Instituto de Ciências Básicas da Saúde, Universidade Federal do Rio Grande do Sul (UFRGS), Porto Alegre 90035-003, RS, Brazil; layaneayres@gmail.com (L.R.A.); helena.schroeder@hotmail.com (H.T.S.); mari.krasbr@gmail.com (M.K.B.R.); lucaskowalewski7@gmail.com (L.S.K.); jeronimocoelho24@hotmail.com (J.d.R.C.); nath-vargas@hotmail.com (N.d.S.V.); pauloivo@ufrgs.br (P.I.H.d.B.J.); 2Programa de Pós-Graduação em Ciências do Movimento Humano, Escola de Educação Física, Fisioterapia e Dança (ESEFID), Universidade Federal do Rio Grande do Sul (UFRGS), Porto Alegre 90690-200, RS, Brazil; 00009478@ufrgs.br; 3Comparative Endocrinology and Metabolism Laboratory (LAMEC), Department of Physiology, Institute of Basic Health Sciences (ICBS), Federal University of Rio Grande do Sul (UFRGS), Porto Alegre 90035-003, RS, Brazil; evvogt@gmail.com (É.L.V.); maizavondentz@hotmail.com (M.C.V.D.); debora.santosrocha@hotmail.com (D.S.R.); jorgefamodel@gmail.com (J.F.A.M.); samirks@hotmail.com (S.K.d.S.); vvitoriagirelli@gmail.com (V.d.O.G.); anapaula.vinagre@ufrgs.br (A.S.V.); 4Department of Sport, Exercise and Rehabilitation, Faculty of Health and Life Sciences, Northumbria University, Newcastle upon Tyne NE1 8ST, UK; eurico.neto@northumbria.ac.uk

**Keywords:** fasting, exercise, heat shock protein, sedentary behavior

## Abstract

**Background:** Physical inactivity induces insulin resistance (IR) and metabolic imbalances before any significant changes in adiposity. Recent studies suggest that the beneficial effects of exercise can be potentiated if performed while fasting. This work aimed to compare the subacute effects of fed- and fasted-state single-bout exercise on biochemical parameters and cellular signaling in the metabolism. **Methods:** The animals were allocated into fed rest (FER), fasting rest (FAR), fed exercise (FEE), and fasting exercise (FAE) groups. The exercise protocol was a 30 min treadmill session at 60% of $\dot{V}$O_2max_. The fasting groups fasted for 8 h before exercise and were killed after 12 h post-exercise. **Results:** Soleus glycogen concentration increased only in the fasting groups, whereas the triglyceride (TGL) content increased in brown adipose tissue (BAT) and liver in the FAE. The FAE showed decreased plasma total cholesterol concentration compared withthe FAR group. Immunocontent of HSP70, SIRT1, UCP-1, and PGC1-α did not change in any tissue investigated. **Conclusions:** Our results indicate that physical exercise while fasting can have beneficial metabolic effects on sedentary animals. Remarkably, in the FAE group, there was a reduction in total plasma cholesterol and an increase in the capacity of BAT to metabolize and store nutrients in the form of TGLs.

## 1. Introduction

Sedentary lifestyle and physical inactivity are associated with the development of several metabolic disorders such as obesity, insulin resistance (IR), and type 2 diabetes mellitus (T2DM) [1,2]. Increased sedentary behaviors are related to central obesity, impaired circulating TGL concentration, fasting insulin, and oral glucose tolerance test responses [3,4,5,6,7]. When physical inactivity is combined with overfeeding in healthy young men, insulin sensitivity dramatically reduces in a very short period, even without significant changes in adiposity, indicating the importance of skeletal muscle contraction and activity for the maintenance of insulin signaling [8].

Changes in body composition, particularly an increase in visceral white adipose tissue, result in the release of cytokines known to cause low-grade inflammation, oxidative stress, and negative cellular changes that result in metabolic dysfunction in skeletal muscle, liver, adipose tissue, blood vessels, and others tissues, causing IR [9,10]. Increased concentrations of free fatty acid arecommonplace in conditions linked to IR, followed by excessive fat accumulation, consequently preventing glucose uptake in the muscle and raising glucose levels in the liver. Insulin resistance and pancreatic b-cell dysfunction lead to the development of T2DM [11,12]. Given the relationship between obesity, insulin resistance, and inflammation, alternative therapies (non-pharmacological/non-chirurgical) to treat and prevent these conditions are essential to reduce morbidity, improve quality of life, and reduce costs for all related complications induced by metabolic diseases [11].

Physical exercise can bring about several metabolic benefits by altering gene transcription related to myogenesis, carbohydrate metabolism, lipid mobilization, substrate uptake, oxidation, mitochondrial biogenesis, oxidative phosphorylation, and other metabolic responses [13]. Recent studies suggest that the beneficial effects of exercise can be potentiated if performed while fasting, thus promoting additional effects, in comparison to the fed state [14]. Previous research has demonstrated that both increased plasma long-chain fatty acid oxidation and intramuscular triacylglycerol utilization are essential in supporting the increased fat oxidation observed during exercise in the overnight fasted state, at least in lean individuals [15]. In addition to the positive metabolic changes promoted by fasting and exercise, other vital pathways that may explain their potential beneficial effects are related to inflammatory signaling. Resolution of inflammation plays a central role in insulin sensitivity and metabolism [16]. The heat shock response (HSR) is a molecular pathway that promotes insulin sensitivity and decreases inflammation [17] and involves key upstream factors such as the NAD+-dependent deacetylase sirtuin-1 (SIRT1), which is activated by the increase in the NAD+/NADH ratio, induced by metabolic stress (e.g., fasting and exercise) [18]. SIRT1 facilitates the activation of the heat shock factor 1 (HSF-1), enhancing the transcription of molecular chaperones, especially the heat shock protein 70 family (HSP70) [19], a classical molecular chaperon that induces cytoprotection and promotes proteostasis [20]. In addition, this protein exerts a powerful anti-inflammatory effect that is attributed to its interaction and inhibition of the critical inflammatory regulator NF-kB [16].

It was recently demonstrated that exercise performed in the fasted state in obese insulin-resistant rats led to transient positive metabolic changes in fat metabolism that lasted for up to 12 h, particularly reducing blood triglycerides and total cholesterol (both markers of cardiovascular risk) [21]. Herein, the current study aimed to evaluate the subacute effects of a single bout of moderate-intensity exercise performed in a fed or fastedstate on biochemical parameters and cellular signaling involved in metabolism, inflammation, and thermogenesis in sedentary non-obese male Wistar rats. We chose to study the 12 h subacute effects since most of the research using similar interventions analyzed the changes that occurred during the exercise protocol, immediately after, and up to 3 h only. Since some metabolic effects of exercise are known to last for several hours, we aimed to test whether exercise/fasting could induce effects several hours after its cessation.

## 2. Materials and Methods

### 2.1. Animals, Ethics, and Experimental Design

Male Wistar rats, Rattus novergicus (n = 32), 60 days old, were obtained from the Center of Reproduction and Experimentation of Laboratory Animals (CREAL) of Universidade Federal do Rio Grande do Sul (UFRGS). All the facilities used comply with Arouca Law No. 11,794 and are accredited by the National Council for the Control of Animal Experimentation (CONCEA). The animals were maintained (five per cage) in standard conditions: 12 h of light/dark cycle, with controlled temperature (21 ± 2 °C) and relative humidity (70%)with food and water ad libitum. This project was approved by the Ethics Committee on the Use of Animals (CEUA) of UFRGS (under protocol 34271, approval date: 21 March 2018). The number of animals was calculated using the software G*Power version 3.1.9.2 (Schleswig-Holstein, Germany), considering an error probability of α = 0.05 and statistical power of 0.80, considering results from Cozer et al. [22]. In addition, results obtained in a previous study using the same experimental design were also considered [21]. The estimated total sample size obtained was 32 animals.

According to Figure 1, the animals were randomly distributed to one of the four experimental groups: fed rest (FER), fed exercise (FEE), fasted rest (FAR), and fasted exercise (FAE). All animals performed one week of acclimatization to the treadmill, which consisted of five daily 15 min sessions at very low speeds and 0° treadmill grade. A washout period of 5 days between the last session of acclimatization and the acute exercise bout was included to avoid any adaptation or any subacute effects of the last acclimation session. Animals unable to perform exercise would be excluded and replaced (in our case, all animals were included and the results included for all variables analyzed).

Before the main experimental session, animals in the fasting groups (i.e., FAR and FAE) were submitted to a fasting period of 8 h, while those allocated to the FER and FEE groups received food and water ad libitum [23]. In the experimental session, the exercise groups (i.e., FAE and FEE) ran for 30 min on the treadmill with a speed and inclination corresponding to 60% of $\dot{V}$O_2max_ (10 m/min, 0° treadmill grade), according to Rodrigues et al. [24], whereas the rest groups (FAR and FER) remained in their cages for the respective amount of time. All groups then received food and water ad libitum and were euthanized 12 h after the experimental session.

The animals were killed by decapitation, and blood, liver, skeletal muscle (soleus and gastrocnemius), and BAT samples were collected. Blood samples were centrifuged (10 min, 4 °C, 1510× *g*), and the plasma was used to evaluate glucose, total proteins, total cholesterol, lactate, and TGL concentration. Glycogen, lactate, and TGL concentrations were also determined in the tissue samples. HSP70, GAPDH, and SIRT1 protein immunocontent levels were determined in skeletal muscle samples, liver, and BAT. In addition, considering the potential role of BAT in thermogenesis, protein expression of uncoupling protein 1 (UCP-1), peroxisome proliferator-activated receptor-gamma coactivator-1-alpha (PGC-1α), and α-tubulin were also measured in this tissue.

### 2.2. Metabolic Measurements

Glucose, lactate, total proteins, total cholesterol, and TGL in the plasma were measured using enzymatic assay kits (Labtest Diagnóstica SA, Lagoa Santa, Brazil) for spectrophotometer analysis. The glucose, triglycerides, lactate, and cholesterol concentrations were expressed as mg·dL^−1^ of plasma and total proteins as g·dL^−1^.The concentration of glycogen in the tissue samples was determined as previously described [25]. For glucose and lactate measurements, samples were collected in tubes previously treated with 0.1 M NaF to block glycolysis. For glucose analysis, the enzymatic assay was based on the glucose oxidase method. To extract triglycerides, tissue samples were homogenized with 0.9% saline in a ratio of 10:1 (1 mg of tissue to 10 μL of saline), and the concentration was measured according to the kit manufacturer’s protocol [25].

### 2.3. Protein Quantification and Western Blotting

Tissue samples were homogenized (still frozen) in 0.1% (*w*/*v*) SDS buffer containing protease and phosphatase inhibitor cocktail (Sigma Aldrich, Milwaukee, WI, USA) consisting of leupeptin (4.2 μM), aprotinin (0.31 μM), TLCK (N-tosyl-L-lysine chloromethyl ketone, hydrochloride; 20 μM), PMSF (phenyl-methyl-sulfonyl fluoride, 100 μM), sodium orthovanadate (Na_3_VO_4_; 1 mM), sodium molybdate (Na_2_MoO_4_; 1 mM), and b-glycerophosphate (1 mM). Then, the homogenates were centrifuged at 15,000× *g* for 5 min at room temperature, and the supernatant fractions were separated for protein determination. Protein quantification was determined using the BCA Protein Assay Kit (Thermo Scientific, Waltham, MA, USA).

For protein separation, SDS-PAGE (polyacrylamide gel electrophoresis with sodium dodecyl sulfate, mini-PROTEAN^®^ 3 Electrophoresis Cell, BioRad, Hercules, CA, USA) was used with a polyacrylamide concentration of 10%. Approximately 30 µg of the protein extracted from the samples was incubated with Laemmli solution and added to each gel well for electrophoresis. The NC membranes were incubated with one of the primary antibodies: anti-HSP70 1:1000 (Sigma Aldrich, Milwaukee, WI, USA; H5147, dilution 1:1000); anti-SIRT1 (Sigma Aldrich 2501994, dilution 1:500); anti-GAPDH (Sigma Aldrich G9545, dilution 1:1000); anti- UCP-1 (Sigma Aldrich A0545, dilution 1:15,000); anti- PGC-1α (Life Technologies do Brasil BS-1832R, dilution 1:1000), and anti-α-tubulin (Invitrogen Thermo Fisher MA180017, dilution 1:1000), for at least sixteen hours at 4 °C, under constant agitation. After incubation, the membranes were washed with TTBS (1%) and then incubated with one of the appropriated secondary antibodies: anti-mouse peroxidase HRP (Sigma Aldrich A9044, dilution 1:10,000); anti-goat (Sigma Aldrich B7014, dilution 1:50,000); anti-rabbit (Sigma Aldrich A0545, dilution 1:10,000), anti-goat (Sigma Aldrich B7014, dilution 1:50,000), for two hours at room temperature. Membranes were washed with Tris Buffered Saline (TBS) and incubated in a dark room with a chemiluminescence solution. Visualization of the blots was performed using ECL™ Select Western Blotting Detection Reagent (GERPN2106, Sigma Aldrich) in a Video Documentation System (ImageQuant™ LAS 4000) and processed using the software Image Quant TL 7.0 for Windows. Optical densitometry and the bands were quantified using ImageJ (version 1.51f; NIH, Maryland City, MD, USA). The immunocontents of HSP70 and SIRT1 were normalized in terms of GAPDH expression, while for PGC-1α and UCP-1, the normalization was performed using α-tubulin expression. The results were expressed in arbitrary units (AUs).

## 3. Statistical Analysis

The data were first analyzed using the Kolmogorov–Smirnov normality test to check the data distribution and, subsequently, using Levene’s homogeneity test. Normally distributed data were analyzed using two-way ANOVA to compare the effects of fasting (fed × fasting) and exercise (rest vs. exercise) and the interaction between these factors, with Bonferroni’s post hoc test for homogeneous data or Games–Howell for non-homogeneous data. Non-parametric data were examined with the Kruskal–Wallis non-parametric test, complemented by Dunn’s post-test. The study assumed α = 0.05 and analyses were performed with the Statistical Package for Social Sciences (SPSS version 25.0, IBM, Armonk, NY, USA). Graphs were generated through the Graph Pad Prism for Windows (version 8.0.1, Graph Pad Software).

## 4. Results

### 4.1. Blood and Tissue Metabolism

The animals’plasma glucose concentration (Figure 2A) showed no difference between the fasting (*p* = 0.440) and exercise (*p* = 0.065) protocols, nor for the combination of fasting and exercise protocols (*p* = 0.303). The concentration of TGL (Figure 2B) at the end of the study was similar between groups (*p* = 0.278, *p* = 0.615, and *p* = 0.214, for comparison of fasting, exercise, and the two protocols, respectively). However, fasting and exercise reduced the total plasma cholesterol concentration (*p* = 0.023 and *p* = 0.005, respectively), but the combination of exercise and fasting did not lead to further difference (Figure 2C). The concentration of total proteins (Figure 2D) did not show a significant difference between the fasting and exercise groups (*p* = 0.877 and *p* = 0.618, respectively) and fasting + exercise (*p* = 0.110). Plasma lactate concentration (Figure 2E) increased in the fasting condition (*p* = 0.001), while the interaction of exercise and fasting was not significant.

The concentration of glycogen in tissues 12 h after an aerobic exercise session in a fasted or fed state is reported in Figure 3. Compared to the fed state, fasting increased the concentration of glycogen in the soleus muscle (Figure 3B) (*p* = 0.039) and in the heart (E) (*p* = 0.002), while the other tissues did not show a significant difference. Fasting also increased triglycerides in the liver (*p* = 0.025, Figure 4C), and fasting exercise increased the concentration of triglycerides in brown adipose tissue (*p* = 0.010, Figure 4D), respectively. There were no significant changes in the gastrocnemius and soleus muscles.

### 4.2. Cellular Signaling

The immunocontent of heat shock proteins HSP70 and SIRT1 did not present differences in any of the tissues, the soleus and gastrocnemius muscles, liver, or brown adipose tissue, as demonstrated below in Figure 5, Figure 6 and Figure 7, respectively. UCP-1 and PGC-1α also showed no significant difference, as shown in Figure 7.

## 5. Discussion

In this study, we found that both exercise and fasting resulted in positive metabolic effects in sedentary rats, particularly in lipid metabolism by reducing plasma cholesterol levels. This effect lasted for up to 12 h after exercise cessation. However, the combination of fasting and exercise did not appear to produce an additive effect. In addition, fasting animals presented higher levels of lactate, increased muscle and heart glycogen, and increased triglycerides in the liver. These results were not accompanied by significant changes in the expression of SIRT1 and HSP70 in any of the tissues analyzed (liver, brown adipose tissue, gastrocnemius, and soleus muscles), norfor UCP1 and PGC1-α in BAT, at least during the period of recovery studied. The only significant result that the association of exercise and fasting induced was the higher content of triglycerides in the BAT.

Physical inactivity promotes insulin resistance and metabolic imbalances even before any changes in body adiposity are noticeable [8]. Thus, physical exercise is an important element for attenuating or reducing development of metabolic disorders and associated complications. The combination of nutritional strategies with exercise may potentiate its beneficial effects [26]. We recently demonstrated the effects of exercise in the fasted state in an animal model of obesity and insulin resistance [21]. In this work, we investigated whether the metabolic improvements found in obese animals would be present in sedentary animals without a significant increase in adiposity.

It has been widely demonstrated that many diseases are related to a sedentary lifestyle, including obesity and insulin resistance [8,27]. The latter is the trigger of several chronic conditions, such as dyslipidemia, T2DM, endothelial dysfunction, hypertension, and kidney disease, among others [8,27]. Although obese and IR individuals have reduced muscle oxidative capacity, regular physical exercise can, at least partially, improve their metabolic profile and mitochondrial efficiency [28,29]. More recently, exercising in a fasted state has gained popularity since it is a condition that brings about metabolic challenges similar to exercise. The central hypothesis for carrying out such a practice is that fasting could enhance the metabolic effects, particularly on lipids, induced by exercise [30,31,32,33,34].

Considering the potential effect of the association of exercise and fasting on lipid metabolism, our results did not demonstrate significant differences in skeletal muscle triglyceride content 12 h after exercise cessation. It is worth noting that, in our experiment, the animals performed exercise for 30 min and then returned to their cages for a 12 h recovery period, with free access to water and food. Therefore, it is possible that any changes induced by fasting or exercise returned to their original values during the recovery period. Interestingly, in obese rats immediately after intervention, in the soleus, a predominantly oxidative muscle, both fasting and exercise resulted in increased glucose and palmitate oxidation; however, no additional effect was associated with exercise and fasting [21]. That result was accompanied by increased AMPK gene expression [21]. Therefore, it is possible to speculate that the immediate change in palmitate oxidation in the soleus was, at least in part, linked to the subacute effect (12 h later) of fasting and exercise on total plasma cholesterol levels, which were significantly reduced (Figure 2C). This effect, considering cholesterol as a marker of cardiovascular risk, is a promising result for improving the health of sedentary individuals, even more so as it is an effect that lasts for up to 12 h. This cholesterol-lowering effect and plasma triglycerides could also be observed in insulin-resistant obese animals subjected to the same intervention protocols (fasting and exercise) [21].

Fasting and exercise may have resulted in a glycogen-sparing effect in the soleus muscle, possibly associated with increased palmitate oxidation [35]. Our results indicate that fasting or then exercise significantly increased muscle glycogen content in the soleus (Figure 3B). A previous study involving exercise while fasting in humans demonstrated a three-fold increase in muscle glycogen resynthesis in type I muscle fibers during recovery from fasting exercise compared with exercise performed in the fed state [34], corroborating our results. In the same study, the authors found reductions in intramuscular triglyceride (soleus) content in participants who exercised while fasting and greater activation of AMPK.

Linked to the above, another critical factor to consider is hormonal change resulting from exercise and fasting. Although similar measurements were not performed in our study, De Bock et al. (2005) reported higher plasma insulin concentrations during the recovery period in individuals who performed exercise in the fasted state. This response may also be related to greater muscle glycogen reserve from fasting and exercise [34].

It is essential to consider that all the responses discussed so far are related to acute and subacute effects of exercise. Chronic (training) adaptations must be interpreted separately. Although it seems plausible that, considering the apparent advantages of exercising while fasting, training in the fasted state could induce more metabolic adaptations (aerobic/oxidative) than training in the fed state, this has not yet been demonstrated. Studies investigating the effect of training in the fasted or fed state essentially found that carbohydrate intake during physical training had little or no influence on metabolic adaptations to physical training [36]. Although the authors found a decrease in exercise-induced glycogen breakdown and an increase in proteins involved in fat handling after training while fasting [36], fat oxidation during exercise with or without carbohydrate intake was not altered. In addition, the results from these studies were obtained from healthy, trained individuals, and so far, no data are available from sedentary subjects or people with metabolic diseases.

In addition to the metabolic changes promoted by our two types of intervention (fasting and exercise), other changes induced can also relate to inflammatory signalling. One is the heat shock response (HSR), a molecular pathway that promotes insulin sensitivity and decreases inflammation. In our study, the expression of SIRT1 and HSP70 proteins, at least during the recovery period, did not change in any of the tissues analyzed (Figure 5, Figure 6 and Figure 7).

Interestingly, works published in the past decade have demonstrated that machinery activating this pathway is attenuated in patients with inflammatory disorders/diseases, such as obesity, diabetes, and others [37]. It has been suggested that the induction of HSP70 expression (or restoration of the cells’ capacity to induce the HSR) through exercise, metabolic stress (such as fasting), or heat therapy may be a valuable strategy to promote metabolic improvements in these patients [38,39]. The lack of response in HSP70 immunocontent following the recovery period studied may be related to the analysis time after metabolic stress, fasting, or exercise. Several studies have demonstrated that the highest expression of HSP70 after stress, regardless of the tissue, occurs between 4–8 h later [40,41].This is probably due to the peak activation of this protein in the face of metabolic challenges, such as exercise and fasting.

Several pieces of evidence show that changes in the NAD+/NADH ratio increase SIRT1 activity 3 h after metabolic stress [12,41]. Therefore, we suggest that changes in SIRT1 immunocontent and its actions, at least in lean sedentary rats, occurred before 12 h post-exercise/fasting. Furthermore, considering that the animals had free access to food during the recovery period, cellular energy levels had already returned to normal without the need to keep this signaling pathway active. As a result, the HSR pathway and consequently, the expression of HSP70 would be similar to the baseline/rest condition. The observed results in our group of sedentary (but non-obese) rats differed from those obtained in obese IR-rats, where SIRT1 and HSP70 were still increased [21] after the recovery period. As demonstrated for other key metabolic regulators, such as AMPK (which is also an activator of SIRT1 and HSR), in people with metabolic diseases, the induction of this protein is delayed [41,42] after the metabolic challenge.

An interesting result in the present work was the increased triglyceride content in the BAT of animals that performed exercise while fasting (Figure 4D). The primary function of BAT is to dissipate energy in the form of heat, a property driven by the presence of the mitochondrial protein UCP1 that uncouples mitochondrial respiration [43]. The BAT is also densely innervated by the sympathetic nervous system (SNS) and is highly vascularized. The thermogenic capacity of BAT may be necessary for heat production in newborns, essential for hibernating rodents and mammals, and possibly assists in the oxidation of excess dietary energy consumption. BAT dissipates energy as heat to maintain optimal thermogenesis and contribute to energy expenditure in rodents and possibly humans. The energetic processes carried out by BAT require a readily available fuel supply, including glucose and fatty acids. Fatty acids are made available by cellular uptake, de novo lipogenesis, and multilocular lipid droplets in brown adipocytes. BAT also has a high glucose uptake and metabolism capacity and can regulate insulin sensitivity. These properties make BAT an attractive target for treating obesity, diabetes, and other metabolic disorders. Previous experiments using cold-exposed mice have shown that genes involved in glucose metabolism, lipogenesis, FA uptake, and catabolism are upregulated in cold adaptation, and fatty acids are utilized for UCP1 activation [44]. Physical exercise, which requires energy consumption for maintenance, can also stimulate thermogenesis, making the mitochondrial machinery more efficient for metabolizing lipids [45]. Thermogenesis can also be stimulated by hormones such as irisin [45], which is known to induce adipose tissue browning. The fact that we observed an increase in the concentration of triglycerides in the BAT may indicate that performing physical exercise while fasting intensified the uptake of fatty acids and carbohydrates and, possibly, their subsequent use in BAT, a desirable effect in people with obesity and diabetes. However, this result must be interpreted with caution since higher triglyceride concentration in BAT does not necessarily determine an increase in thermogenesis. For example, the increase in the expression of proteins, such as UCP1, does not directly mean an increase in adipose tissue thermogenesis. A recent study found that the immunocontent of UCP1, CPT (carnitine-palmitoyl transferase 1B), and PGC-1α, which could be viewed as an indication of increased thermogenesis activity, was increased by a high-fat diet [46]. However, the high-fat diet does not favor thermogenesis in the tissue because there is a different distribution in the storage of triglycerides, concentrated in larger lipid droplets, which makes access to lipases difficult, favoring, in fact, storage rather than the use and oxidation of fat. This result is in line with the results obtained in our study, since the healthy animals in the study by Da Eira et al. [46] also increased the quantity of BAT triglycerides, but in this case, favoring its use in thermogenesis.

The responses in BAT to training are still conflicting, where training can result in both increasing or attenuating the capacity of BAT to promote thermogenesis [47]. In a study that evaluated acute and chronic exercise responses in rats acclimatized to different environmental temperatures, the group that underwent training decreased the content of UCP-1 in BAT, but the expression of PGC-1α was increased [47]. In another study, the group that performed acute exercise until exhaustion showed increased expression of UCP-1 and PGC-1α [48]. It is worth noting that the observed responses were seen immediately after the exercise (acute), which differed from our work, where the analysis was carried out 12 h after the end of the exercise session (subacute). During exercise, muscle contraction generates heat, and these higher temperatures could attenuate the thermogenesis generated by BAT in response to exercise [49]. On the other hand, a recently published work demonstrated that the induction of a local temperature increase in adipose tissue resulted in increased thermogenesis in rodents and humans. However, this effect was exclusive to beige adipose tissue, as brown adipose tissue does not respond to increases in temperature [50]. High-intensity resistance training markedly reduces PGC-1α and UCP-1 content in adipose tissue, suppressing fatty acid oxidation [51]. It is possible that training generates adaptations so that circulating lipids are directed to the muscle. In other words, BAT may reduce its activity in response to training due to decreased circulating fatty acids and cholesterol. BAT is highly dependent on lipids for thermogenesis [52]. This consideration may be important, since fasting can favor the activity of BAT by providing greater metabolization and the release of lipids into circulation [53].

In sedentary rats, BAT may offer protective effects, both through thermogenesis and the uptake of lipids, even if these are not used to generate heat. However, these responses seem to be limited to moments close to exercise. It is still necessary to evaluate subacute responses in other populations with more significant metabolic impairment, such as people with obesity, insulin resistance, and diabetes. Finally, to determine the capacity of exercise and fasting to induce thermogenesis, different time points of evaluation are necessary, from immediately after the metabolic challenge to the following minutes and hours. In addition, other proteins and mitochondrial activity are now being evaluated in our laboratory.

Finally, our work has limitations that are important to consider. Firstly, we only analyzed single time points after the exercise/fasting intervention (12 h after) and free fatty acids were not measured. It will be essential to study different times to understand the behavior and time course of protein expression/metabolic adaptations (e.g., immediately after, 1, 2, 3, 6, 12, and 24 h after). In addition, $\dot{V}$O_2max_ was not directly assessed. Thus, exercise intensity may only have been precise for some exercised animals despite our control using venous blood lactate concentration (to guarantee that exercise was aerobic/not intense).

## 6. Conclusions

Our results indicate that performing physical exercise while fasting can result in beneficial metabolic effects for sedentary individuals. Remarkably, there was a reduction in total plasma cholesterol and an increase in the capacity of brown adipose tissue to metabolize and store nutrients in the form of triglycerides. The fact that we observed an increase in the concentration of triglycerides in the BAT and liver may indicate that performing physical exercise while fasting intensified the uptake of fatty acids and carbohydrates and, possibly, their subsequent use. This is a desirable effect in people with obesity, IR, and diabetes. In conclusion, it is essential to consider the potential metabolic benefits of exercise with fasting for sedentary individuals, as it can improve lipid metabolism and overall health. However, further research is needed to fully understand the long-term effects and specific adaptations that may result from this combination.

## Figures and Tables

**Figure 1 nutrients-16-03529-f001:**
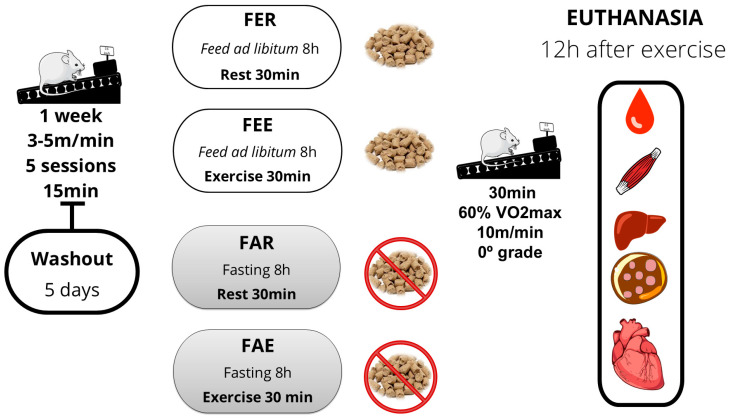
Experimental design. Groups: fed rest (FER), fed exercise (FEE), fasted rest (FAR), and fasted exercise (FAE).

**Figure 2 nutrients-16-03529-f002:**
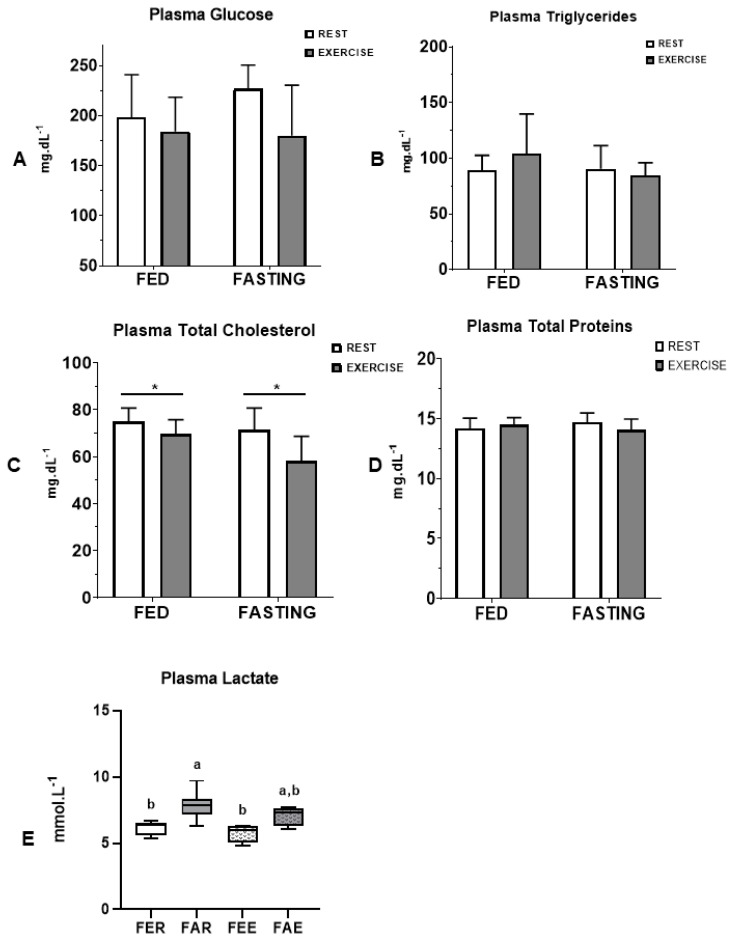
Blood and tissue metabolism. Concentrationsof plasma glucose (**A**), TGL (**B**), total cholesterol (**C**), total proteins (**D**), and lactate (**E**) are presented.Groups: fed rest (FER), fed exercise (FEE), fasted rest (FAR), and fasted exercise (FAE). Data expressed as mean ± SD. * represents differences between groups. Letters represent significant differences between each experimental group.

**Figure 3 nutrients-16-03529-f003:**
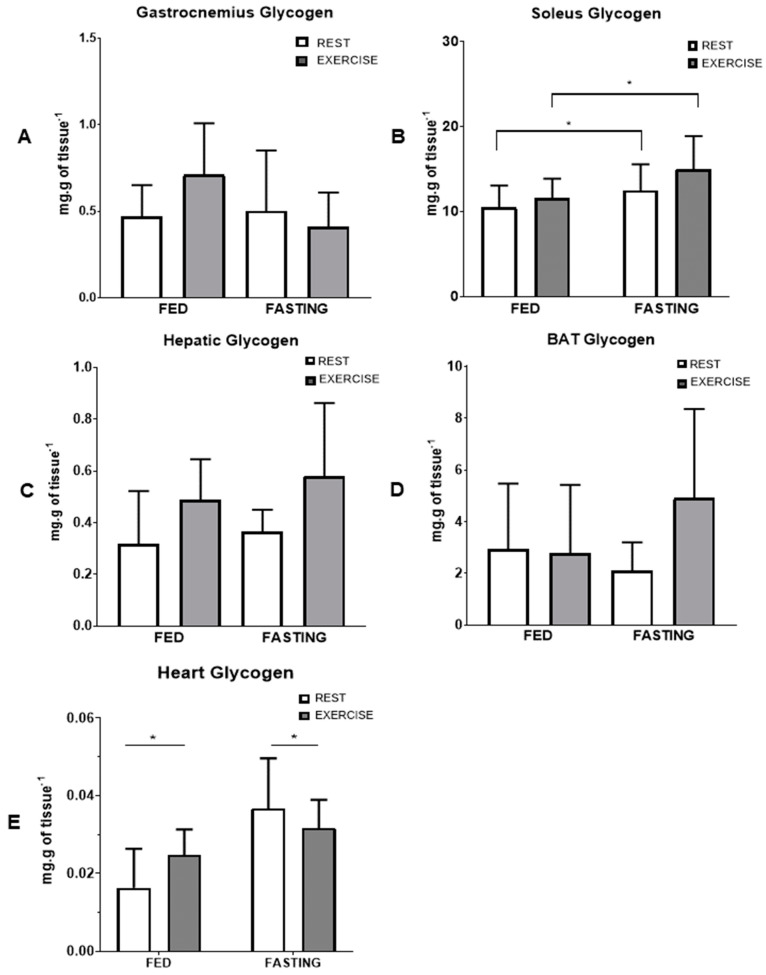
Glycogen concentration in the tissues in the final experimental protocol. Gastrocnemius muscle (**A**), soleus muscle (**B**), liver (**C**), brown adipose tissue (**D**), and heart (**E**) glycogen concentrations are presented. Data expressed as mean ± SD. * represents differences between groups.

**Figure 4 nutrients-16-03529-f004:**
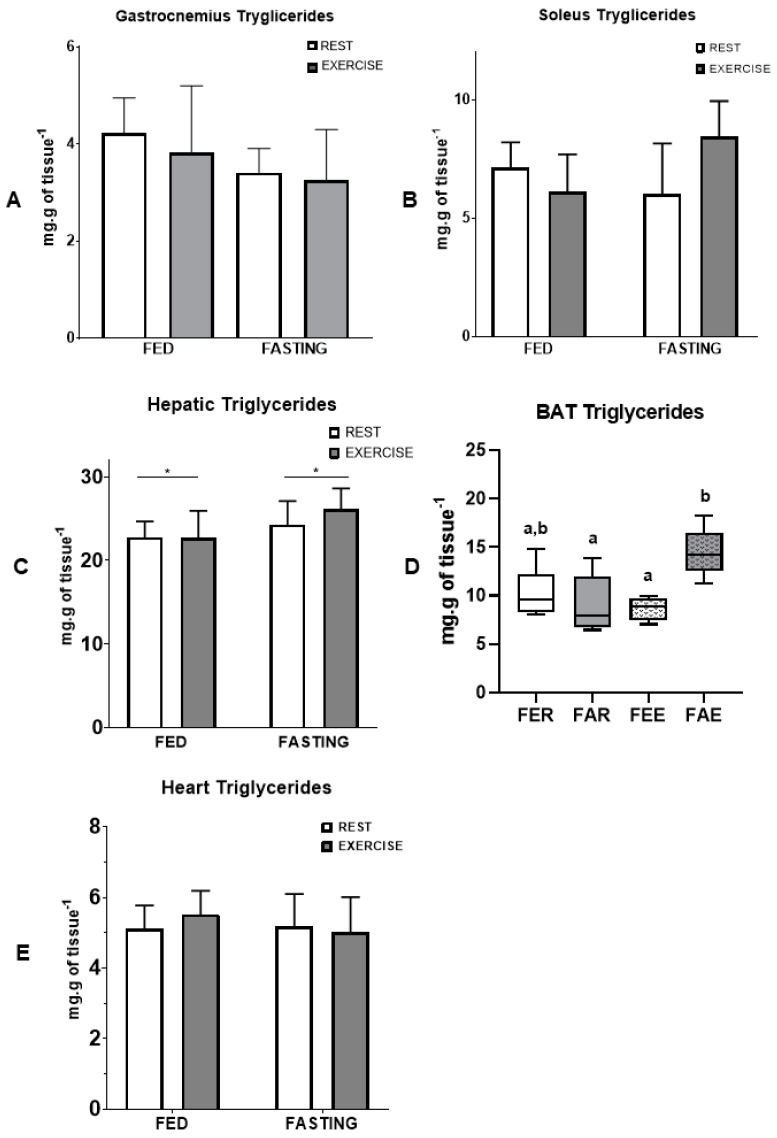
Concentration of triglycerides in the tissues in the end experimental protocol. Gastrocnemius muscle (**A**), soleus muscle (**B**), liver (**C**), brown adipose tissue (**D**), heart (**E**). Data expressed as mean ± SD. * represents differences between groups. Letters represent significant differences between each experimental group.

**Figure 5 nutrients-16-03529-f005:**
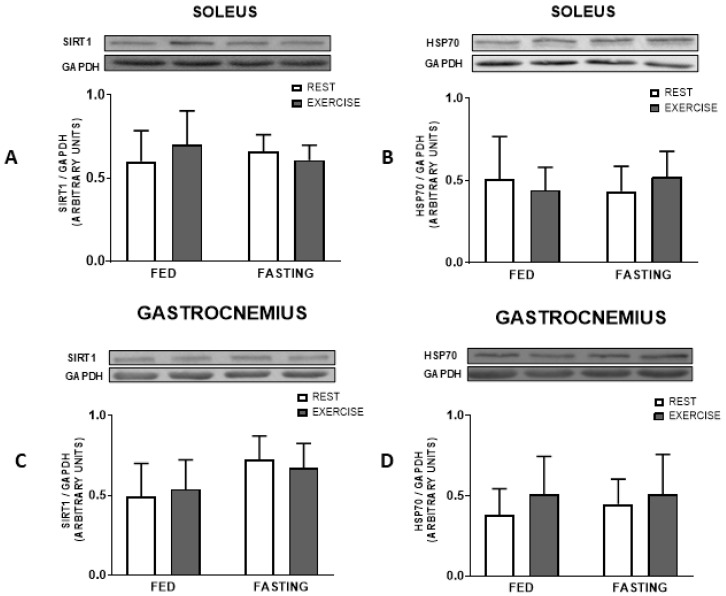
Protein immunocontent at 12 h post-experimental treatment. SIRT1 and HSP70 in the soleus (**A**,**B**) and gastrocnemius muscle (**C**,**D**), respectively. Data expressed as mean ± SD.

**Figure 6 nutrients-16-03529-f006:**
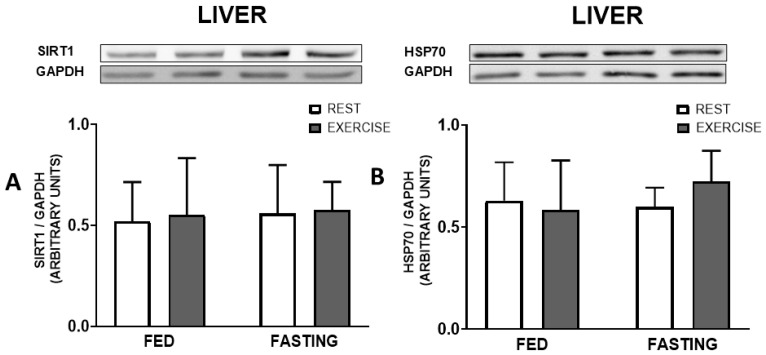
Protein immunocontent at 12 h post-experimental treatment. SIRT1(**A**) and HSP70 (**B**) in the liver. Data expressed as mean ± SD.

**Figure 7 nutrients-16-03529-f007:**
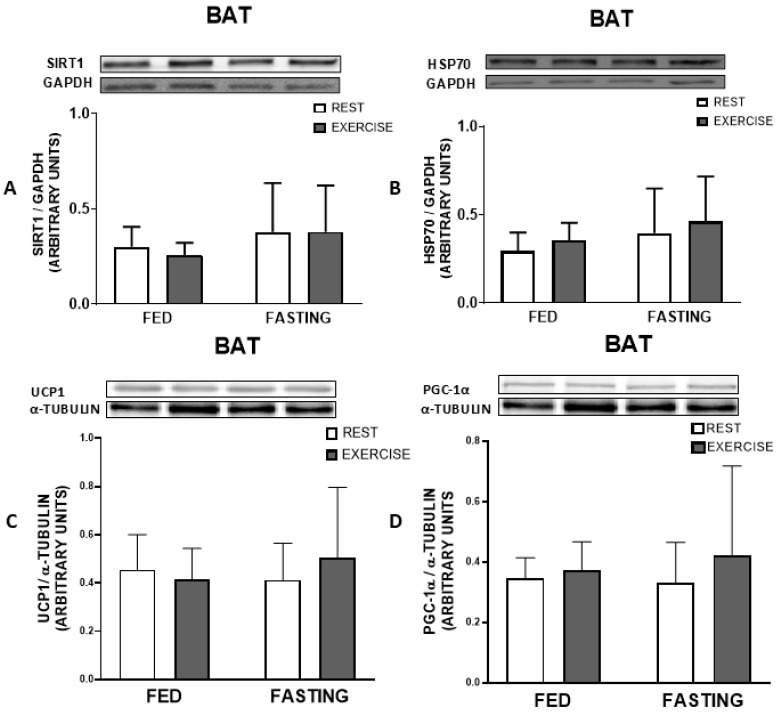
Protein immunocontent at 12 h post-experimental treatment. SIRT1 (**A**), HSP70 (**B**), UCP1 (**C**), PGC-1α (**D**) in BAT. Data expressed as mean ± SD.

## Data Availability

Data available on request from the authors.

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
