# Peer review of "Subacute Effects of Moderate-Intensity Aerobic Exercise in the Fasted State on Cell Metabolism and Signaling in Sedentary Rats"

_nutrients, 2024, doi:10.3390/nu16203529_

Round 1

Reviewer 1 Report

Comments and Suggestions for Authors

Layane Ramos Ayres and colleagues measured biochemical parameters and cellular signaling pathways in plasma and multiple organs of obese mice. They examined the subacute effects of a single bout of exercise performed in a fed or fasted state. Their findings suggest that fasting exercise can have beneficial metabolic effects on sedentary animals. This was indicated by a reduction in total plasma cholesterol and an increase in the capacity of brown adipose tissue (BAT) to metabolize and store nutrients as triglycerides.

Addressing Concerns

1.      Comparison to the IJERPH Study:

There is one paper published in July 2021 in the International Journal of Environmental Research and Public Health (IJERPH). The title is: "Metabolic and Molecular Subacute Effects of a Single Moderate-Intensity Exercise Bout, Performed in the Fasted State, in Obese Male Rats."

 The study design and results are highly similar. Could you clarify the differences between this study and the one I mentioned earlier?  

2.      Metabolic Effects of Acute, Subacute, and Chronic Exercise:

In this study, the authors performed a single exercise session in obese mice. Could you discuss the metabolic effects of acute, subacute, and chronic exercise?

Author Response

Layane Ramos Ayres and colleagues measured biochemical parameters and cellular signaling pathways in plasma and multiple organs of obese mice. They examined the subacute effects of a single bout of exercise performed in a fed or fasted state. Their findings suggest that fasting exercise can have beneficial metabolic effects on sedentary animals. This was indicated by a reduction in total plasma cholesterol and an increase in the capacity of brown adipose tissue (BAT) to metabolize and store nutrients as triglycerides.”

Addressing Concerns

  1. Comparison to the IJERPH Study:

There is one paper published in July 2021 in the International Journal of Environmental Research and Public Health (IJERPH). The title is: "Metabolic and Molecular Subacute Effects of a Single Moderate-Intensity Exercise Bout, Performed in the Fasted State, in Obese Male Rats."

 The study design and results are highly similar. Could you clarify the differences between this study and the one I mentioned earlier? 

  1. Metabolic Effects of Acute, Subacute, and Chronic Exercise:

In this study, the authors performed a single exercise session in obese mice. Could you discuss the metabolic effects of acute, subacute, and chronic exercise?

Author Response: We thank the reviewer for the positive observation regarding our study. As you will see, in response to all reviewers’ comments, we undertook a comprehensive effort to enhance the quality of our findings and manuscript. We hope that our changes demonstrate the thoroughness of our study and are satisfactory to you.

The IJERPH study, published in 2021, was conducted using the same type of intervention (acute exercise and fasting). The difference is that in the previous work, we tested the effects of acute exercise and fasting in diet-induced obese / insulin-resistant rats. Since we observed some positive metabolic and molecular changes, we decided to test whether these changes would also be found in a less severe metabolic-compromised animal, in this case, sedentary (but not obese / nor insulin resistant) rats. This new work found similar blood metabolic changes but not the same tissue molecular adaptations (protein expression). For example, both animal models have shown that exercise and fasting reduced the total cholesterol concentrations. However, in obese/insulin-resistant animals (IJERPH study), additional changes were found, such as lower plasma triglycerides and higher expression of skeletal muscle proteins (HSP70 and SIRT1).

In the present work, we aim to analyze only the subacute effects of a single exercise or fasting intervention. We chose to study the subacute since most of the research using similar interventions analyzed the changes that occurred during the exercise protocol, immediately after, and up to 3 hours only (Vogt et al., 2023; Edinburgh et al., 2020). Since some metabolic effects of exercise are known to last for several hours (for example, glucose uptake/insulin sensitivity, van Dijk JW et al. 2013), we aimed to test if exercise/fasting could induce effects several hours after its cessation. We understand that the fact that we analyze only 12 hours after the intervention is a limitation of our work. Therefore, we added a sentence including this limitation to the discussion.

References:

Vogt ÉL, Von Dentz MC, Rocha DS, Model JFA, Kowalewski LS, Silveira D, de Amaral M, de Bittencourt Júnior PIH, Kucharski LC, Krause M, Vinagre AS. Acute effects of a single moderate-intensity exercise bout performed in fast or fed states on cell metabolism and signaling: Comparison between lean and obese rats. Life Sci. 2023 Feb 15;315:121357. doi: 10.1016/j.lfs.2022.121357. Epub 2023 Jan 10.

Edinburgh et al. Lipid metabolism links nutrient-exercise timing to insulin sensitivity

in men classified as overweight or obese. J Clin Endocrinol Metab, Estados Unidos,

  1. 105, n. 3, p. 660-676, mar. 2020.

Van Dijk JW, et al. Effect of moderate-intensity exercise versus activities of daily living on 24-hour blood glucose homeostasis in male patients with type 2 diabetes.

Reviewer 2 Report

Comments and Suggestions for Authors

Review of the manuscript: nutrients-3161465

 Subacute effects of moderate-intensity aerobic exercise, in the fast state, on cell metabolism and signaling in sedentary rats

 The aim of the study was to evaluate the subacute effects of a single bout of moderate-intensity exercise performed in a fed or fasted-state on biochemical parameters and cellular signaling involved in metabolism, inflammation, and thermogenesis in sedentary non-obese male Wistar rats.”

 The subject of the study is very interesting, since there are some studies suggesting additional beneficial effects of exercise performed during fasting comparing to exercise in the fed state.  

 In the Introduction section the Authors accurately justified the purpose of the study, based on a good knowledge of current literature.

The study design and the applied methods of research are proper and well described.

Statistical analysis is properly applied and results are well described.

 The Authors indicted that performing physical exercise during fasting condition has beneficial metabolic effects in sedentary rats – a reduction in total plasma cholesterol in serum and an increase in the capacity of brown adipose tissue to store and metabolize triglyceroides. In light of the obtained results, they concluded that „it is essential to consider the potential metabolic benefits of exercise with fasting for sedentary individuals”.

 The manuscript is well written, however I have a few remarks:

1) The Authors should explain why they were marking GAPDH in the tissue samples?

2) Line 198 is p=0.216 correct?

3) Line 231 – there is an “e” – is this correct?

4) Line 331 and 332 – it looks like something is missing,

5) Line 347 – Figure 4c or 4d?

Author Response

“The aim of the study was “to evaluate the subacute effects of a single bout of moderate-intensity exercise performed in a fed or fasted-state on biochemical parameters and cellular signaling involved in metabolism, inflammation, and thermogenesis in sedentary non-obese male Wistar rats.”

 The subject of the study is very interesting, since there are some studies suggesting additional beneficial effects of exercise performed during fasting comparing to exercise in the fed state. 

 In the Introduction section the Authors accurately justified the purpose of the study, based on a good knowledge of current literature.

The study design and the applied methods of research are proper and well described.

Statistical analysis is properly applied and results are well described.

 The Authors indicted that performing physical exercise during fasting condition has beneficial metabolic effects in sedentary rats – a reduction in total plasma cholesterol in serum and an increase in the capacity of brown adipose tissue to store and metabolize triglyceroides. In light of the obtained results, they concluded that „it is essential to consider the potential metabolic benefits of exercise with fasting for sedentary individuals”.

Author Response: We thank the reviewer for the positive observation regarding our study and the generated results. As you will see, in response to all reviewers’ comments, we did considerable work to improve the quality of our findings and manuscript. We hope that our changes are satisfactory.

 The manuscript is well written, however I have a few remarks:

1) The Authors should explain why they were marking GAPDH in the tissue samples?

Author Response: We used GAPDH and tubulin as normalizer proteins for calculation (Protein of interest/normalizer), as required for western blot. This information was added to the text as follows:

“The immunocontents of HSP70 and SIRT1 were normalized in terms of GAPDH expression, while for PGC-1α and UCP-1, the normalization was performed using α-tubulin expression. The results were expressed in arbitrary units (AU).”

2) Line 198 is p=0.216 correct?

3) Line 231 – there is an “e” – is this correct?

4) Line 331 and 332 – it looks like something is missing,

5) Line 347 – Figure 4c or 4d?

Author Response: All the required changes were made.

Reviewer 3 Report

Comments and Suggestions for Authors

A sedentary and lacking physical activity lifestyle is indeed associated with many metabolic diseases. The focusing point of this manuscript is good, but there are some problems in the design and writing. 

Prior to significant changes in obesity, lack of exercise could induce insulin resistance, but this study did not conduct tests on glucose tolerance and insulin tolerance. But also, this study did not conduct obesity modeling. The purpose mentioned in the preface is to reduce blood triglycerides and total cholesterol after 12 hours of exercise in obese rats in a fasting state. However, in this study, the rats were normal and their triglycerides and total cholesterol levels should be normal. Why some indices will be changes after 12 hours of exercise and fasting? The authors did not provide a good interpretation. 

Animals are unable to perform stationary or inactive interventions. Is this statement appropriate? 

How the exercise mode, intensity, and duration determined? Why choose this mode? The authors attached a reference [25] which were used for the diabetic rats, so the 10m/min of rate which was not equal to 60% VO2max for normal rats. The authors should test the max rate or the VO2max for the rat. 

Animals were forced to exercise, but cannot be equated with prolonged sitting like humans, they constantly moving up and down, and move around, and the term 'sedentary rats' cannot be used to “rest”. So the rest groups remained in the cages, but were unsure to stop moving.

 According to the ethical requirements in nowadays, the fasting time for rodent experiment cannot exceed 6 hours. However, in this study, a fasting period of 8 hours does not comply with for rodent experimental animals in FER and FEE.

 Figure 1, under the mouse in the picture, "1 weak?" Figure 1 is drawn relatively roughly.

 In line 198, Figure 2C shows significant differences between exercise and fasting, but P=0.216; Figure 2E: Is there a significant increase in plasma lactate concentration under fasting conditions? What do a and b represent in Figure 2E?

What is the normal serum lactate concentration for rats. If it is greater than 2 mmol/L, it indicates hyperprolactinemia. In this study, the normal lactate concentration was written as 6mmol/L, while the fasting lactate concentration was 8mmol/L. After exercise, lactate concentration will increase, but in this study it decreased. How to explain this?

 In this study, heart glycogen increased after exercise, increased glycogen in soleus muscle, whether fasting or not. The explanation is that the oxidation of palmitate oxidation has increased in Line 290. Has this data been detected?

What do a and b represent in Figure 4D? Why does it display differently from other images? Suggest using a scatter bar chart to represent.

 After 12 hours of exercise, there was no change in SIRT1 protein levels, and the author speculated that the change occurred before 12 hours post-exercise/fasting, but there is no reference.

 This study only observed changes in indicators after 12 hours intervention and did not observe changes immediately, 1 hour, 2 hours, and 6 hours after exercise, lacking temporal consistency.

 Last but not least, this study lacks morphological results to prove some issues. The protein expression chose to detect by WB, without significantly, cannot explain the purpose of this study.

Author Response

“A sedentary and lacking physical activity lifestyle is indeed associated with many metabolic diseases. The focusing point of this manuscript is good, but there are some problems in the design and writing.“

Author Response: We thank the reviewer for the positive observation regarding our study and the generated results. As you will see, in response to these comments, we did considerable work to improve the quality of our findings and manuscript.

1) Prior to significant changes in obesity, lack of exercise could induce insulin resistance, but this study did not conduct tests on glucose tolerance and insulin tolerance. But also, this study did not conduct obesity modeling. The purpose mentioned in the preface is to reduce blood triglycerides and total cholesterol after 12 hours of exercise in obese rats in a fasting state. However, in this study, the rats were normal and their triglycerides and total cholesterol levels should be normal. Why some indices will be changes after 12 hours of exercise and fasting? The authors did not provide a good interpretation.

Author Response: We understand the reviewer's point. However, in any model (human or animal), subjects who do not perform regular exercise, in our and many other researchers' opinion (Hongyang et al., 2018; Batacan et al., 2016; Holland et al.), are not healthy. Laboratory rats are sedentary because they are housed in conditions where activity is limited. Thus, compared with animals with access to exercise (voluntary or forced), sedentary animals present higher levels of total cholesterol and triglycerides, for example (Hongyang et al., 2018; Batacan et al., 2016; Holland et al.). For this reason, interventions such as exercise and fasting may reflect changes over these parameters that, as we demonstrated, indeed occur. In addition, as we mentioned in the introduction, we previously tested the same intervention in an animal model of obesity and insulin resistance, finding similar (and even more significant) results (Vogt et al. 2021 and 2022). We hope that the reviewer understands our point of view.

References:

Physiological and biochemical characteristics of skeletal muscles in sedentary and active rats Hongyang Xu 1, Xiaoyu Ren, Graham D Lamb, Robyn M Murphy . 2018 Apr;39(1-2):1-16. doi: 10.1007/s10974-018-9493-0. Epub 2018 Jun 15. J Muscle Res Cell Motil.

Batacan, R. B., Duncan, M. J., Dalbo, V. J., Connolly, K. J., & Fenning, A. S. (2016). Light-intensity and high-intensity interval training improve cardiometabolic health in rats. Applied Physiology, Nutrition, and Metabolism, 41(9), 945–952. doi:10.1139/apnm-2016-0037.

Holland AM, Kephart WC, Mumford PW, Mobley CB, Lowery RP, Shake JJ, Patel RK, Healy JC, McCullough DJ, Kluess HA, Huggins KW, Kavazis AN, Wilson JM, Roberts MD. Effects of a ketogenic diet on adipose tissue, liver, and serum biomarkers in sedentary rats and rats that exercised via resisted voluntary wheel running. Am J Physiol Regul Integr Comp Physiol. 2016 Aug 1;311(2):R337-51.

Vogt ÉL, Von Dentz MC, Rocha DS, Argenta Model JF, Kowalewski LS, de Souza SK, Girelli VO, de Bittencourt PIH Jr, Friedman R, Krause M, Vinagre AS. Metabolic and Molecular Subacute Effects of a Single Moderate-Intensity Exercise Bout, Performed in the Fasted State, in Obese Male Rats. Int J Environ Res Public Health. 2021 Jul 15;18(14):7543.

Vogt ÉL, Von Dentz MC, Rocha DS, Model JFA, Kowalewski LS, Silveira D, de Amaral M, de Bittencourt Júnior PIH, Kucharski LC, Krause M, Vinagre AS. Acute effects of a single moderate-intensity exercise bout performed in fast or fed states on cell metabolism and signaling: Comparison between lean and obese rats. Life Sci. 2023 Feb 15;315:121357. doi: 10.1016/j.lfs.2022.121357. Epub 2023 Jan 10.

2) Animals are unable to perform stationary or inactive interventions. Is this statement appropriate?

Author Response: We needed to understand the reviewer's question. This statement is not present in our manuscript

3) How the exercise mode, intensity, and duration determined? Why choose this mode? The authors attached a reference [25] which were used for the diabetic rats, so the 10m/min of rate which was not equal to 60% VO2max for normal rats. The authors should test the max rate or the VO2max for the rat.

Author Response: We understand the reviewer's concern. Unfortunately, we cannot directly measure VO2 max in animals (rats). The reason why we chose reference 25 (diabetic rats) is that our first research using this intervention was in an animal model of diet-induced obesity and insulin resistance. Considering the positive metabolic and molecular findings from this study, we decided to investigate if, in a less severe metabolic model, prior to obesity (in our case, sedentary animals), the benefits would be similar. For this reason, we apply the same protocol. We understand that exercise intensity may vary without this assessment. However, to guarantee that our exercise was aerobic (below the anaerobic threshold), we measured, during the sessions, the venous lactate concentration (from the tail) before and after the exercise. Exercise increased the lactate concentration; however, it remains below the anaerobic threshold (4 mmol/L), indicating that our exercise was performed at an aerobic intensity, which was our main proposal. We added this as a limitation of the work as follows:

“Finally, our work has limitations that are important to consider. Firstly, we only analyzed one-time points after the exercise/fasting intervention (12 hours after). It would be essential to study different times to understand the behavior and time course of protein expression/ metabolic adaptations (ex, immediately after, 1, 2, 3, 6, 12, and 24 hours after). In addition, VO2 max was not directly assessed. Thus, exercise intensity may only be precise for some exercised animals despite our control using venous blood lactate concentration (to guarantee that exercise was aerobic/not intense).”

4) Animals were forced to exercise, but cannot be equated with prolonged sitting like humans, they constantly moving up and down, and move around, and the term 'sedentary rats' cannot be used to “rest”. So the rest groups remained in the cages, but were unsure to stop moving.

Author Response: Yes, we agree with the reviewer's observation that the animals may move constantly in the cage. We discussed the term “sedentary” previously (response to comment 1). The term “rest” was used only to differentiate from the animal that performed the exercise on the treadmill for 30 minutes. The term, in these conditions (exercise vs. not exercise), is widely used, including in our previous publications.  

5) According to the ethical requirements in nowadays, the fasting time for rodent experiment cannot exceed 6 hours. However, in this study, a fasting period of 8 hours does not comply with for rodent experimental animals in FER and FEE.

Author Response: We acknowledge the reviewer's concern, but we firmly believe that the 8-hour fasting period is crucial for our study's objectives.

As reviewed and described in detail by Secor and Carrey (2016), each animal has a characteristic cycle of feeding and digestion followed by an episode of interprandial fasting. If feeding is frequent, such fasting bouts are relatively brief, during which the animals experience only a modest fluctuation in energy flux and homeostasis (Secor & Carrey, 2016). The animal must endure extended fasting bouts with a compounding negative energy budget (Secor & Carrey, 2016). Therefore, the fasting state can be divided into 3 phases, which vary in duration according to each species' daily routine. Phase I of fasting is relatively short and encompasses the transition from meal digestion and assimilation to digestive quiescence and the employment of fasting responses. This phase is accompanied by an initial decline in metabolic rate as the energy-consuming meal digestion and assimilation processes are suspended. The source of metabolized substrate switches from the previous meal to complete reliance upon body stores of glycogen, lipids, and protein. Endotherms may nearly deplete glycogen stores during phase I to maintain glucose availability. Phase II is dominated by lipid utilization, while protein catabolism usually starts in Phase III (Secor & Carrey, 2016).

As described in detail by Andrade Jr. (2017), the human daily routine includes three primary meals, breakfast, lunch, and dinner, which are interspersed by fasting periods of approximately five hours each, and the most extended fasting period corresponds to nocturnal fasting (around eight hours). Fasting metabolism in lean subjects is characterized by low insulin levels, high glucagon levels, liver glycogenolysis, and gluconeogenesis for maintaining serum glucose levels and cerebral function. It involves high levels of lipolysis and FFA via circulating TAG to enable energy utilization by tissues other than the brain and the central nervous system (CNS). If the fasting period is more extended, progressive ketosis develops due to the mobilization and ß-oxidation of fatty acids and the increase in ketone bodies (KBs) (e.g., ß-hydroxybutyrate) that replace glucose as the primary energy source in the CNS, thereby decreasing the need for gluconeogenesis and sparing protein catabolism (Andrade Jr, 2017).

Our decision to implement an 8-hour fasting period was based on the need to simulate the conditions under which people typically exercise and fast. Since we did not determine the exact moment when the rats began feeding, we considered 8 hours of fasting to be when the rats are at the end of Phase I or transitioning to Phase II. A longer fasting period would likely lead to the depletion of glycogen reserves and a shift from lipid catabolism to protein breakdown, potentially compromising the rats' ability to sustain aerobic exercise.  

References:

Stephen M Secor , Hannah V Carey. Integrative Physiology of Fasting Compr Physiol. 2016 Mar 15;6(2):773-825. doi: 10.1002/cphy.c150013.

Andrade Jr. Metabolism during Fasting and Starvation: Understanding the Basics to Glimpse New Boundaries. Journal of Nutrition and Dietetics, 2017.

6) Figure 1, under the mouse in the picture, "1 weak?" Figure 1 is drawn relatively roughly.

Author Response: Figure 1 was modified accordingly.

7)  In line 198, Figure 2C shows significant differences between exercise and fasting, but P=0.216; Figure 2E: Is there a significant increase in plasma lactate concentration under fasting conditions? What do a and b represent in Figure 2E?

Author Response: The difference (*) is related to rest vs. exercise. Regarding lactate, fasting resulted in higher lactate levels. A and b represent the differences between the groups.

8) What is the normal serum lactate concentration for rats. If it is greater than 2 mmol/L, it indicates hyperprolactinemia. In this study, the normal lactate concentration was written as 6mmol/L, while the fasting lactate concentration was 8mmol/L. After exercise, lactate concentration will increase, but in this study it decreased. How to explain this?

Author Response: As we mentioned before (question 3, for exercise intensity), we measure venous lactate (from the animal tail) as a marker for exercise intensity (widely used). In contrast, at the end of the protocol, the blood, for general metabolites, was collected after decapitation. The venous lactate concentration at rest was 2.2± 0.5 mmol/L and increased to 3±0.4 following exercise (data not shown). The results described in the manuscript (collected after decapitation) present higher lactate concentrations. This is explained by the fact that with decapitation, other sources of lactate can be mixed, such as from the cerebrospinal fluid and arterial blood. Lactate concentration is higher in cerebrospinal fluid than in the blood and increases more in response to low-to-moderate exercise (E Bisgard et al.). Lactate in the cerebrospinal fluid is already higher at rest (≥4mmol/L) and increases to 7mmol/L during moderate exercise. We believe that this is the explanation for the higher level of lactate. In this case, the result does not represent any physiological hyperprolactinemia. Finally, it is essential to remember that our results represent the body's metabolism 12 hours after the exercise. Thus, it would not be expected to find lactate levels elevated 12 hours after exercise.

Reference

E Bisgard, H V Forster, B Byrnes, K Stanek, J Klein, M Manohar. Cerebrospinal fluid acid-base balance during muscular exercise. J Appl Physiol Respir Environ Exerc Physiol. 1978 Jul;45(1):94-101. doi: 10.1152/jappl.1978.45.1.94.

9)  In this study, heart glycogen increased after exercise, increased glycogen in soleus muscle, whether fasting or not. The explanation is that the oxidation of palmitate oxidation has increased in Line 290. Has this data been detected?

Author Response: In previous work, we measured the acute effects of the same type of intervention; sedentary obese rats increased soleus palmitate oxidation (Vogt et al., 2022). This result suggests that the previous increase in palmitate oxidation, which may have occurred acutely, could reflect a glycogen-sparing effect. Previous studies already described similar findings using sedentary rats (G.L. Dohm et al., 1983). Since this is more speculative, we added this information to the text (discussion). We hope that this change is satisfactory.

References

Vogt ÉL, Von Dentz MC, Rocha DS, Model JFA, Kowalewski LS, Silveira D, de Amaral M, de Bittencourt Júnior PIH, Kucharski LC, Krause M, Vinagre AS. Acute effects of a single moderate-intensity exercise bout performed in fast or fed states on cell metabolism and signaling: Comparison between lean and obese rats. Life Sci. 2023 Feb 15;315:121357. doi: 10.1016/j.lfs.2022.121357. Epub 2023 Jan 10.

G.L. Dohm, E.B. Tapscott, H.A. Barakat, G.J. Kasperek, Influence of fasting on  glycogen depletion in rats during exercise, J. Appl. Physiol. Respir. Environ. Exerc.  Physiol. 55 (1983) 830–833.

10) What do a and b represent in Figure 4D? Why does it display differently from other images? Suggest using a scatter bar chart to represent.

Author Response: We thank the reviewer for the suggestion. Indeed, changing the type of graph is more appropriate (nonparametric data).

11) After 12 hours of exercise, there was no change in SIRT1 protein levels, and the author speculated that the change occurred before 12 hours post-exercise/fasting, but there is no reference.

Author Response: We added the reference to the text.

Cantó, C.; Gerhart-hines, Z.; Feige, J.N.; Lagouge, M.; Milne, J.C.; Elliott, P.J.; Puigserver, P.; Auwerx, J. AMPK Regulates Energy Expenditure by Modulating NAD + Metabolism and SIRT1 Activity. Nature2009, 458, 1056–1060.

12) This study only observed changes in indicators after 12 hours intervention and did not observe changes immediately, 1 hour, 2 hours, and 6 hours after exercise, lacking temporal consistency.

Author Response: We agree with the reviewer that understanding the time course of metabolic and molecular changes induced by our intervention would be important. Unfortunately, we don’t have data describing these other time points, which was added as a limitation of our work.

“Finally, our work has limitations that are important to consider. Firstly, we only analyzed one-time points after the exercise/fasting intervention (12 hours after). It would be essential to study different times to understand the behavior and time course of protein expression/ metabolic adaptations (e.g. immediately after, 1, 2, 3, 6, 12, and 24 hours after). In addition, VO2 max was not directly assessed. Thus, exercise intensity may only be precise for some exercised animals despite our control using venous blood lactate concentration (to guarantee that exercise was aerobic/not intense).”

13)  Last but not least, this study lacks morphological results to prove some issues. The protein expression chose to detect by WB, without significantly, cannot explain the purpose of this study.

Author Response: As stated in the introduction, our purpose was to investigate the subacute effects of exercise and fasting over some molecular/metabolic pathways that we understand are important for metabolism and inflammatory control (SIRT, HSP70). We chose western blot analysis since the effects were measured hours after the exercise, allowing enough time to induce protein expression. Unfortunately, for these animals (sedentary), after 12 hours, no significant effect was found. We understand that this is also attributed to our choice of analysis 12 hours after the exercise cessation, which is a limitation of our work. As previously mentioned, we added this as a limitation in the discussion.  

Round 2

Reviewer 3 Report

Comments and Suggestions for Authors

Thanks for the response and revision of the manuscript by the authors. There are still some mistakes in it. For example, what are the meaning of  a and b stand for in Fig2E, Fig 4D? The WB bands should be put upper in Fig 5-7,  and the Histogram should be under the bands.

Author Response

“Thanks for the response and revision of the manuscript by the authors. There are still some mistakes in it. For example, what are the meaning of  a and b stand for in Fig2E, Fig 4D? The WB bands should be put upper in Fig 5-7,  and the Histogram should be under the bands. “

Author Response: We thank the reviewer for the final suggestions. All the required changes were made.